# The Effect of Vaccination with Live Attenuated Neethling Lumpy Skin Disease Vaccine on Milk Production and Mortality—An Analysis of 77 Dairy Farms in Israel

**DOI:** 10.3390/vaccines8020324

**Published:** 2020-06-19

**Authors:** Michal Morgenstern, Eyal Klement

**Affiliations:** Koret School of Veterinary Medicine, Hebrew University of Jerusalem, Rehovot 76100, Israel; michal.morgenstern@mail.huji.ac.il

**Keywords:** Neethling, vaccination, milk loss, mortality rate, adverse effects, field-controlled, survival analysis, lumpy skin disease

## Abstract

Lumpy skin disease (LSD) is an economically important, arthropod borne viral disease of cattle. Vaccination by the live attenuated homologous Neethling vaccine was shown as the most efficient measure for controlling LSD. However, adverse effects due to vaccination were never quantified in a controlled field study. The aim of this study was to quantify the milk production loss and mortality due to vaccination against LSD. Daily milk production, as well as culling and mortality, were retrieved for 21,844 cows accommodated in 77 dairy cattle farms in Israel. Adjusted milk production was calculated for each day during the 30 days post vaccination. This was compared to the preceding month by fitting mixed effects linear models. Culling and mortality rates were compared between the 60 days periods prior and post vaccination, by survival analysis. The results of the models indicate no significant change in milk production during the 30 days post vaccination period. No difference was observed between the pre- and post-vaccination periods in routine culling, as well as in immediate culling and in-farm mortality. We conclude that adverse effects due to Neethling vaccination are negligible.

## 1. Introduction

Vaccines are among the most important measures taken for control of infectious diseases, both in human and veterinary medicine. However, vaccination can cause adverse effects, which can result in morbidity and economical losses. Obviously, this varies between different diseases and different vaccines. Decision-making regarding vaccination, therefore depends on accurate data that should enable appropriate weighing of the pros of vaccines against their cons, as a measure for prevention of specific diseases.

Lumpy skin disease (LSD) is a vector-borne viral disease, which primarily affects cattle. It is caused by the lumpy skin disease virus (LSDV), a member of the Poxviridae family and the genus Capripoxvirus [1]. The disease is mostly characterized by the occurrence of localized or generalized skin nodules. It is often accompanied by lethargy, reduced appetite, edema, reduction in milk production, and might even cause death [1]. Reduction in milk production and mortality are considered to be among the most important causes for LSD-associated direct economic losses in dairy cattle, though they were never quantified.

Until 2012, LSD was mainly limited to Africa with some sporadic incursions which caused outbreaks in Israel during 1989, 2006, and 2007 [2,3,4,5] and, according to the World Organization for Animal Health (OIE), in Kuwait (1991), Lebanon (1993), Yemen (1995), United Arab Emirates (2000), Bahrain (2003), and Oman (2010) [1]. In 2012, a large epidemic occurred in Israel, which lasted until August 2013 [6]. In 2014, the virus occurred in Turkey and caused a large epidemic, which was not fully controlled to date [7]. This was followed by spread of LSD to Greece, during 2015 and to Bulgaria and other Balkan countries, a year later [8], causing a vast epidemic that was eventually controlled two years later [9]. During the last years, LSD had spread to Armenia, Azerbaijan, Kazakhstan, Georgia, and the Russian Federation [10,11] and was recently diagnosed in China, India, Israel, Bangladesh, and Syria [12].

Several attenuated vaccines are used to date against LSD and generated by passaging isolated viruses serially in tissue culture and/or eggs until attenuation is achieved [13]. These vaccines can be divided to heterologous vaccines, based either on Sheep-pox or Goat-pox virus or homologous vaccines based on attenuated LSD virus [1]. There are currently three companies in South Africa that produce LSDV homologous vaccines. Two companies are using vaccines containing cell-adapted strains of the original LSDV Neethling strain (LSD Vaccine for Cattle, Onderstepoort Biological Products; OBP, South Africa and Bovivax, MCI Santè Animale, Morocco). The third company is using an attenuated South African LSDV field isolate (Lumpyvax, MSD Animal Health-Intervet, South Africa) [7,11]. Some challenge studies showed the efficacy of heterologous (mainly goat-pox based) vaccines [14,15]. However, to date, no controlled field study was performed to evaluate their efficacy. The same is true for the homologous Kenyan goat and sheep pox (KGSP) vaccine, which was also associated with severe adverse effects when used in Israel [16]. The only vaccine for which field-controlled data exist is a homologous vaccine, based on an attenuated Neethling LSD strain. This vaccine was successfully used for controlling the LSD epidemics in Israel and the Balkans, where its efficacy (relative to RM65 Sheep-pox live attenuated vaccine administered at the same dose as the Neethling vaccine) and effectiveness were estimated at 76.6% and 79.8% (on average), respectively [6,9]. During the efficacy study conducted in Israel a self-limiting Neethling associated disease was observed in 0.38% of the vaccinated cows [6]. This phenomenon was characterized by the appearance of small nodules up to two weeks after vaccination and was associated with isolation of the Neethling strain from the affected cows. Since then, several research groups reported on adverse effects associated with Neethling vaccination, mainly edema at the injection site, generalized skin nodules, and decrease in milk production [8,17,18,19]. However, these studies were non-controlled and suffered from limited sample size.

Accurate information on vaccine associated adverse effects is prudent for evidence-based decision-making regarding vaccination and possible compensation. We therefore conducted this study on 77 dairy farms in Israel accommodating 21,844 milking cows in order to quantify the effect of vaccination with live attenuated Neethling LSD vaccine on milk production losses, culling, and mortality.

## 2. Materials and Methods

### 2.1. Study Rationale

The study rationale is to compare milk production, mortality, urgent culling, and routine culling in farms vaccinated with the Neethling vaccine, prior and after vaccination. Using data from a multiplicity of farms is expected to overcome potential collinearity of the period (before or after vaccination) with other coincidental events occurring at the farms.

### 2.2. Study Population

The dairy farms in Israel are divided mainly into two types: cooperative farms (kibbutz) and family farms. In 2018, 58% of the milk was produced by cooperative farms, 41% was produced by family farms, and 1% by agriculture school farms. The average size of the cooperative farms in 2018 was 474 cows, with an average annual milk production of 5,826,000 kg/farm. The average size of the family farms at that year was 119 cows, and the average annual milk production was 1,361,000 kg/farm. All cows in the study were vaccinated by the veterinarians of the “Hachaklait” organization, which provides veterinary medicine services to more than 80% of the Israeli dairy cattle. We screened the “Hachaklait” vaccination database to find farms that vaccinated cattle during 2019. Overall, 116 vaccinating farms were detected, of which data were available on 82. We then excluded farms that experienced significant morbidity cause by other agents. Three farms were excluded due to the occurrence of brucellosis, one due to bovine tuberculosis, and one due to vitamin A deficiency. The study thus included 77 farms of Israeli Holstein cows held under zero grazing in free-stall sheds. Of these, 53 were family farms, 23 were herds of cooperative farms, and one was of an agriculture school farm. Of these, 73 farms were vaccinated routinely for at least five years. Routine vaccination was first performed in calves aged three months and was repeated yearly in all the cattle aged >0.5 years (all the cattle in the farm were vaccinated at the same date). Another four farms were vaccinated on 2019 and 2016 but were not vaccinated during 2017 and 2018. These farms were vaccinated according to the same routine. However, in 2019 cows in lactation 1 and some of the lactation 2 cows were vaccinated for the first time with no prior vaccination at the age of three months. All 77 farms were vaccinated with the live attenuated Neethling LSD vaccine (OBP, South Africa, Lot numbers: 455,463,467) according to the manufacture instructions:

(https://www.obpvaccines.co.za/resources/productInserts/LUMPY%20SKIN%20DISEASE%20VACCINES%20FOR%20CATTLE.pdf).

### 2.3. Data Collection

Data and code for analysis are provided in doi: 10.5281/zenodo.3899468. Daily milk records along with date of calving, lactation group, and date of occurrence of sudden death/urgent culling/and routine culling were provided by the Israeli Cattle Breeders Association. Data were collected for the period from 15 October 2018 until 31 August 2019. None of the farms tested were affected by LSD or other known OIE-listed diseases during the study follow-up period. Date of vaccination was provided by the “Hachaklait” for each farm. The original crude data included 4,720,633 observations/daily milk records.

### 2.4. Data Analysis

#### 2.4.1. Analysis of the Effect of Vaccination on Milk Production

Analysis of milk production data was performed in two stages by fitting a mixed effects linear models to the data. The first stage was performed in order to calculate the daily milk production of each cow adjusted for days in milk (DIM). The adjustment to DIM was performed with reference to three stages in the milk production curve: 1. Sharp increase in milk production after calving; 2. Decrease; 3. Plateau or very mild decrease. In order to define these three stages, we plotted the daily average milk per DIM for lactation group 1, lactation group 2, and lactation group >2 (Appendix A). The following equation was fitted to the daily milk production for each cow for each of the three lactation groups separately:(1)$$Y_{i,j,k}=\beta_{0}+\beta_{1}DIM_{i,j,k}+\beta_{2}H_{i,j,k}+\beta_{3}DIM_{i,j,k}H_{i,j,k}+u_{0j,k}+v_{0k}+e_{i,j,k}$$
where *Y*_*i*,*j*,*k*_ is the daily milk production in day *i*, in cow *j* in farm *k*; *DIM* is the number of days since calving (days in milk); *H* is a nominal variable of DIM with three categories according to the three periods as been defined in Appendix A; *β*_0_ is the intercept; *β*_1_ is the fitted slope of *DIM*; *β*_2_ is the fitted slope of *H*; *β*_3_ is the fitted slope of the interaction of *DIM* and *H*; *u*_0*j*,*k*_ is the random intercept of cow *j* in farm *k*; *v*_0*k*_ is the random intercept of farm *k*; and *e*_*i*,*j*,*k*_ is the residual error term.

The residuals of daily milk were retrieved from the results of this model and were used for further analysis of milk loss. The average of the daily milk production residuals for the 30 days prior vaccination of each cow was subtracted from the daily milk residual, beginning at vaccination day (day 0) until 30 days post vaccination. This milk production gap (MPG) was used to analyze reduction in milk production after vaccination. As detailed in Figure 1, a total of 442,664 observations from 15,002 cows in the 73 routinely vaccinated farms were available for this final analysis (see Figure 1, “Data for Equation (2)”). 

Based on previous data [20], there are several suggested mechanisms underlying adverse reactions to vaccines. The main two mechanisms are: 1. Short-term direct response to the vaccination itself. 2. Development of a disease due to infection by the vaccine attenuated virus. We therefore hypothesized that the effect of vaccine on milk production will be differentiated to four periods based on the day since vaccination (DIV): DIV 0 (at day 0), DIV 1–7 (days 1–7), DIV 8–14, DIV 15–30. In addition, in order to see the overall change during 30 days after vaccination, we fitted another model, including only two DIV categories: DIV 0 and DIV 1–30 (days 1–30).

The two following linear mixed effects models were fitted to the MPG. The model fitted to MPG for each lactation separately is described by Equation (2). The model fitted to MPG for all lactations is described by Equation (3).
(2)$$MPG_{j,k}=\beta_{0}+\beta_{1}DIV_{j,k}+v_{0k}+e_{j,k},$$
(3)$$MPG_{j,k}=\beta_{0}+\beta_{1}DIV_{j,k}+\beta_{2}L_{j,k}+v_{0k}+e_{j,k},$$
where *MPG*_*i*,*j*,*k*_ is the daily milk production gap of cow *j* in farm *k*, *DIV* is the number of days since vaccination (nominal variable), *L* is the lactation group (nominal variable with three categories: lactation group 1, lactation group 2, and lactation group>2), *β*_0_ is the intercept, *β*_1_ is the fitted slope for *DIV*, *β*_2_ is the fitted slope for *L*, *v*_0*k*_ is the random intercept of farm *k*, and *e*_*j*,*k*_ is the residual error term. As mentioned above, for each of the two model formats, three models were fitted in which *DIV* was categorized in three ways: 31 categories (0–30), two categories (0,1–30), and four categories (0,1–7,8–14,15–30).

#### 2.4.2. The Effect of Neethling Vaccination on Culling and Mortality

This effect was examine using survival analysis. The rationale of this analysis is to compare the culling and mortality rate before and after vaccination. We therefore compared between two time periods: 1. Pre-vaccination period—from sixty days prior vaccination until exit or vaccination. 2. Post vaccination period—from vaccination day until exit day or sixty days post vaccination. The number of cows and farms analyzed is described in Figure 2.

##### Time and Event Definition

Separate analysis was performed for two event definitions: 1. Any type of exit from the farm, i.e., death in the farm, urgent culling, and routine culling. 2. Only unexpected culling, i.e., death in the farm and urgent culling. When this second event definition was analyzed routine culling was treated as right censoring. All cows were censored at day 60 of follow-up of the relevant period.

##### Univariable Survival Analysis

Survival comparison between the pre and post vaccination periods was visualized by Kaplan-Meier survival curves and was tested for statistical significance by the log-rank test.

##### Multivariable Survival Analysis

A multivariable Cox proportional hazards model was fitted to the data to control for DIM and lactation. Cows were classified to three groups according to their lactation (as described for the milk production analysis). Lactation was included in the model as a fixed nominal variable. DIM at the day of follow-up onset was defined according to the last calving occurring before the follow-up period. Cows were then divided to four (1–4) categories according to their DIM (0 ≤ DIM ≤ 90, 91 ≤ DIM ≤ 180, 181 ≤ DIM ≤ 270, 271 ≤ DIM, respectively). This DIM category was included as a fixed nominal factor in the model. All two-way interactions were tested for statistical significance.

#### 2.4.3. The Effect of Neethling Vaccination on Milk Production and Culling/Mortality on Naïve Cows

The same procedures were performed separately for the four farms which were not vaccinated in 2017–2018. In these farms, all the cows in lactation 1 were assumed to be vaccinated for the first time, as well as some of the lactation 2 cows, while the >2 lactation cows were assumed to be vaccinated prior to the vaccination occurring study period. This part included 7457 observations from 260 cows: 79 naïve cows (lactation group 1), 110 non-naïve cows (lactation group >2), and 71 naïve and non-naïve cows (lactation group 2) (Figure 1).

Statistical analysis was performed using R version 3.6.0 (R Core Team, Vienna, Austria), the “lme4”, the “survival”, and the “survminer” packages [21,22,23,24]. Ninety-five percent confidence intervals were calculated by using the ‘profile method in ‘confint’ command in R.

## 3. Results

### 3.1. Analysis of Milk Production during 30 Days Following Vaccination

As depicted in Figure 3, and Table 1 no post-vaccination significant decrease in MPG was observed except for a slight decrease at day five. Moreover, at lactation group 1, there is even a slight increase in MPG.

Analysis of milk production following Neethling vaccination on farms with naïve cows shows a similar pattern (Figure 4). Milk production, during the 30 days post vaccination, in lactation groups 1, 2, >2, and combined shows no significant reduction except for day 2 and day 21 after vaccination. Notably, this milk reduction is lowest among the lactation 1 group (i.e., first vaccination naïve group) (Figure 4).

### 3.2. The Effect of Neethling Vaccination on Culling and Mortality

No significant difference in in-farm mortality and immediate culling was demonstrated between the pre and post vaccination periods (Figure 5a, LogRank *p*-value = 0.3). The same results were observed in the multivariable Cox proportional hazards model, after controlling for DIM and lactation group (Table 2). Surprisingly, a significant increase in total survival (including routine culling) was observed after vaccination (Figure 5b, LogRank *p*-value = 0.02). However, it should be noted that the survival curves cross each other, indicating on possible interaction with time. Multivariable analysis could not be performed for this outcome due to failure to comply with the proportional hazards assumption.

Survival analysis was not performed for the previously unvaccinated farms due to small number of culling/mortality events.

## 4. Discussion

This is the first controlled field study and the largest to quantify milk loss and mortality due to vaccination against LSD. The vaccine used in this study (Neethling LSD vaccine, OBP, South Africa) was previously demonstrated to have high efficacy and effectiveness against LSD and was successfully used for controlling the LSD epidemics in Israel and the Balkans [6,9]. The results of this study indicate that there is no significant milk production loss during the 30 days following live-attenuated Neethling vaccination. Moreover, mortality and culling rate are not influenced by vaccination.

No significant milk loss was documented during the entire month period post vaccination. Katsoulos et al. and Bedekovic et al. demonstrated viremia from 5 up to 21 days post vaccination [18,19]. Our findings suggest that such a potential viremia is of no clinical significance. Ben-Gera et al. reported of an adverse event occurring at least two weeks after vaccination, which was named ‘Neethling disease’. However, this phenomenon was reported only in 0.38–0.6% of the vaccinated cows and was mostly of mild nature [6]. Thus, even if it did occur in our study, it failed to reduce overall milk production.

Our findings are opposed to the findings of Abutarbush et al. [25] and Katsoulos et al. [19], who described significant reduction of up to 5.5–16% in milk production for a period of up to several weeks post vaccination. These studies, however, suffer from a very small sample size of only one herd for the quantitative analysis. In addition, these studies do not incorporate variance in the cow level as they are based on milk quantification at the tank level and are not controlled. With the normal daily fluctuations in milk yield and the significant covariates that are not controlled for (e.g., days in milk and lactation group), such methodology cannot be accepted. Furthermore, while Katsoulos et al. studied vaccination by a commercial LSDV vaccine, in Abutarbush et al.’s study, the type of vaccine administered is not known as at that time farmers in Jordan used two vaccines, one of which was unlabeled and was later identified as a strain of LSDV.

Another concern related to vaccination is the occurrence of severe adverse events. These events usually occur in very small numbers both in humans [26,27] and in animals [28]. The current study was designed in order to find these rare events. Two types of events were analyzed: 1. Any type of exit from the herd, i.e., death in the farm, urgent culling, and routine culling. 2. Only unexpected culling, i.e., death and urgent culling. This separate analysis was performed in order to examine vaccination effect, not just mortality, but also on routine culling. Both univariable and multivariable (controlled for DIM and lactation) survival analysis show that there is no significant difference in mortality and routine culling rate between the pre- and post-vaccination periods.

One may argue that incidence of adverse events after vaccination might differ between naïve and previously vaccinated animals [29]. Most of the farmers who vaccinated their herds in 2019, vaccinated them routinely in previous years. Therefore, only four farms were adequate for examining the influence of vaccination on naïve cows. From these four farms, only cows from lactation group 1 were naïve to LSD vaccination with high certainty. The sample size for examining this question is therefore small. However, this analysis shows that milk loss after vaccination in all lactation groups was similarly minimal and short with even less significant change in naïve cows. Interestingly, we also noticed a short and significant milk loss on day 21 post vaccination. It might suggest an adverse event related to clinical infection by the attenuated virus. However, the short duration of the milk production reduction and the fact that it is most prominent in previously vaccinated cows suggest that it might be related to a coincidental occurrence of another stressful event during the post vaccination period. Such an event could incidentally be more significant in the higher lactation groups, thus causing excessive reduction in milk production compared to the first lactation groups. Occurrence of severe adverse events could not be tested in this sample due to the rarity of these events and the small sample size.

### Limitations

Due to the retrospective nature of this study, we could not test directly the immune response elicited by this vaccine. However, evidence for the protective immunity elicited by the used vaccine batches is provided by the fact that in an epidemic, which occurred shortly after the study period, no cases of LSD occurred in vaccinated cows. The main limitation of this study is that it does not deal with specific adverse effects, such as abortions. This was impossible in this study due to wide and unspecific definition of abortion in the Israeli dairy farm. It should be mentioned, however, that the basic argument regarding potential association of abortions with the disease itself or with vaccination is anecdotal and was never proved. A controlled large-scale study to examine this potential association is therefore warranted. Future research could also be performed in order to analyze the influence of vaccination on conception rate. In addition, as we mentioned, the sample size for measuring adverse effects in naïve cows was small. Though it supports no association with previous vaccination, a larger study examining this question is warranted. Lastly, one might argue that the results of the study are relevant only to highly industrialized zero grazing dairy milk farms and not to smaller farms which are sometimes managed poorly. Though this may be a valid claim, we believe that the highly productive cow in industrialized farms is regularly in much higher stress than cattle reared at smaller farms. We therefore suspect that, if no vaccination associated adverse reduction in survival and milk production were observed in the current study, then such adverse effects in smaller farms are even less plausible.

## 5. Conclusions

The current study shows that the adverse effects of Neethling vaccination are minor and negligible. Considering the disease spreading to India and East Asia and the high effectiveness of the Neethling vaccine, the data of this study show that vaccination against LSD by the Neethling attenuated vaccine is recommended without reservations.

## Figures and Tables

**Figure 1 vaccines-08-00324-f001:**
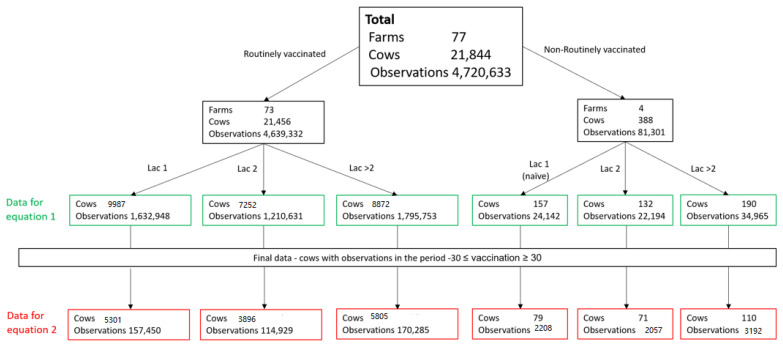
Overall number of observations and cows available for milk production analysis in each lactation group in 77 farms.

**Figure 2 vaccines-08-00324-f002:**
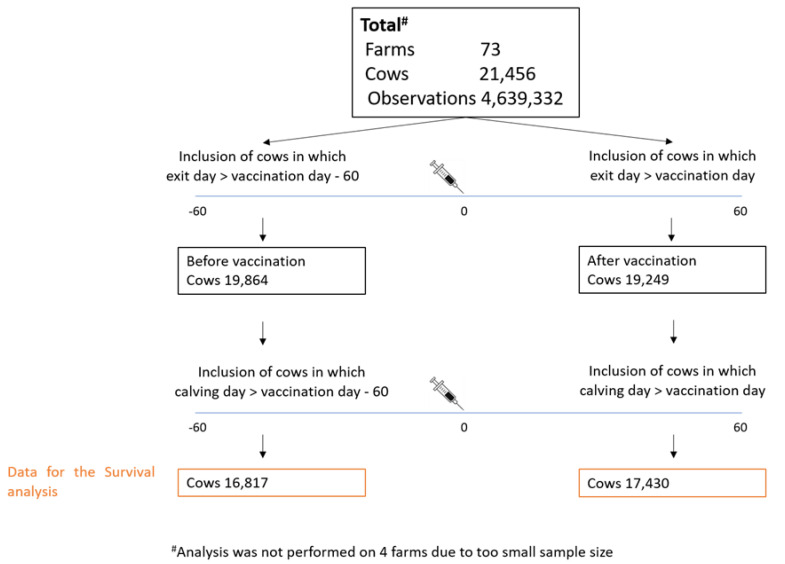
Overall number of observations and cows available for survival analysis in each lactation group in 73 farms.

**Figure 3 vaccines-08-00324-f003:**
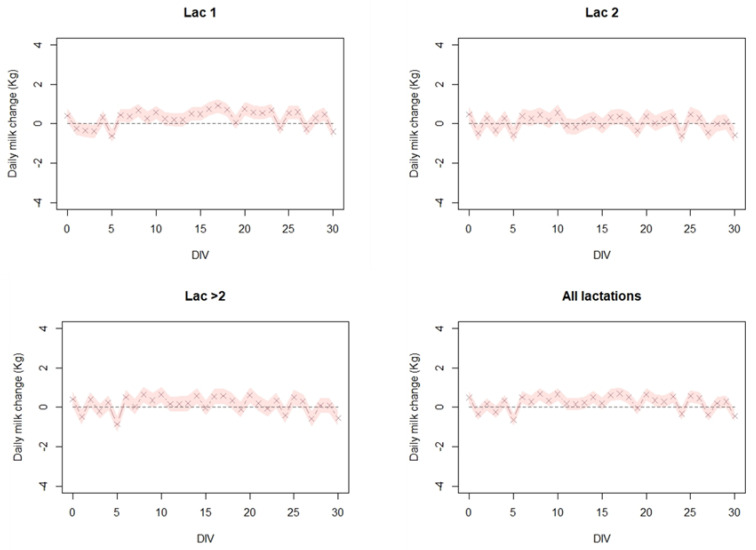
The estimated daily milk change (Kg) per cow in 73 dairy cattle farms in Israel, during a 30-day period after vaccination. Linear mixed effects model was fitted to the milk production gaps (MPG) for each cow with distance from vaccination day (DIV) as a fixed variable and the farm as a random variable. This was performed separately for lactation group 1 (Lac 1), lactation group 2 (Lac 2), lactation group >2 (Lac > 2), and all lactation groups combined (All lactations). For specific details on the model, see text.

**Figure 4 vaccines-08-00324-f004:**
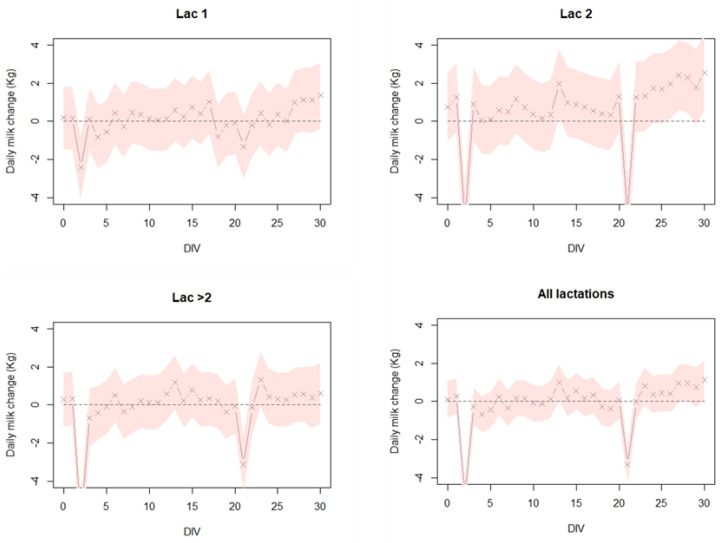
The estimated daily milk change (Kg) per cow in 4 dairy cattle farms in Israel, during a 30-day period after vaccination. Linear mixed effects model was fitted to the milk production gaps (MPG) for each cow with distance from vaccination day (DIV) as a fixed variable and the farm as a random variable. This was performed separately for lactation group 1 (Lac 1-naive), lactation group 2 (Lac 2-mixed: naïve and vaccinated), lactation group >2 (Lac > 2-vaccinated), and all lactations groups combined (All lactations). For specific details on the model, see text.

**Figure 5 vaccines-08-00324-f005:**
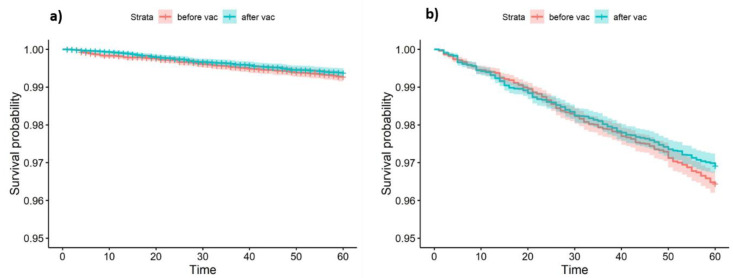
Kaplan-Meier survival curves as a function of vaccination status in 73 dairy cattle farms in Israel. (**a**) Event defined as in-farm death or urgent culling. (**b**) Event defined as in-farm death, urgent culling, or routine culling.

**Table 1 vaccines-08-00324-t001:** Estimated lactation specific daily milk change per cow, in 73 dairy cattle farms in Israel at different time periods after vaccination. See text for details on analysis.

Lactation Group	DIV	Estimate (95% CI)
1	0	0.416 (0.084; 0.748)
	1–7	−0.062 (−0.367; 0.242)
	8–14	0.384 (0.079; 0.689)
	15–30	0.411 (0.109; 0.713)
	0	0.415 (0.083; 0.748)
	1–30	0.291 (−0.010; 0.592)
2	0	0.484 (0.095; 0.874)
	1–7	−0.028 (−0.378; 0.322)
	8–14	0.168 (−0.182; 0.518)
	15–30	0.041 (−0.304; 0.387)
	0	0.484 (0.095; 0.874)
	1–30	0.055 (−0.290; 0.399)
>2	0	0.404 (0.022; 0.787)
	1–7	−0.057 (−0.403; 0.288)
	8–14	0.388 (0.043; 0.734)
	15-30	0.118 (−0.224; 0.459)
	0	0.404 (0.021; 0.787)
	1–30	0.140 (−0.201; 0.480)
All lactations	0	0.490 (0.185; 0.795)
	1–7	0.009 (−0.282; 0.301)
	8–14	0.390 (0.099; 0.681)
	15–30	0.263 (−0.027; 0.553)
	0	0.490 (0.185; 0.795)
	1–30	0.232 (−0.057; 0.522)

**Table 2 vaccines-08-00324-t002:** Multivariable Cox regression analysis results in 73 dairy cattle farms in Israel. Event is defined as in-farm death or urgent culling.

Covariate	HR	*p*-Value	95% CI
Vaccination status ^1^	0.878	0.327	0.677; 1.139
DIM 2 ^2^	0.644	0.033	0.430; 0.965
DIM 3 ^3^	0.555	0.012	0.350; 0.880
DIM 4 ^4^	1.221	0.209	0.894; 1.666
Lactation 2	1.268	0.255	0.842; 1.910
Lactation >2	2.749	1.74 × 10 ^−9^	1.978; 3.820

^1^ Vaccinated ^2^ 91 ≤ DIM ≤ 180 at entry day of follow up ^3^ 181 ≤ DIM ≤ 270 at entry day of follow up ^4^ 271 ≤ DIM at entry day of follow up.

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
