# Peer review of "The Effect of Vaccination with Live Attenuated Neethling Lumpy Skin Disease Vaccine on Milk Production and Mortality—An Analysis of 77 Dairy Farms in Israel"

_vaccines, 2020, doi:10.3390/vaccines8020324_

Round 1
Reviewer 1 Report
Morgenstern and Klement presented the results of a large scale analysis of potential adverse effects of vaccination with a live-attenuated lumpy skin disease vaccine. As the co-authors point out this is an rapidly emerging and poorly studied animal pathogen. The co-authors were not able to identify any significant effect either in milk production or mortality after vaccination. I have no major concerns with this manuscript.
minor comments
l. 105 to avoid confusion please use 15 Oct 2018 and 31 Aug 2019.
l. 305 Please explain more either here or in the discussion why there was a significant decrease only on two days. Do the co-authors think this is due to vaccination or are there other factors that could explain the decrease? Why would the reduction be less in group 1 and more in the others?
l. 432 "East Asia" instead of "far East"
l. 433-434 This concluding statement is a bit weak. "the data show" instead of "we believe". "without reservations" instead of "by all means".
l. 435 Include the appendix as Supplemental Material or delete the Supplementary Materials statement.
Author Response
Morgenstern and Klement presented the results of a large-scale analysis of potential adverse effects of vaccination with a live-attenuated lumpy skin disease vaccine. As the co-authors point out this is an rapidly emerging and poorly studied animal pathogen. The co-authors were not able to identify any significant effect either in milk production or mortality after vaccination. I have no major concerns with this manuscript.
Au: We thank the reviewer for his support.
minor comments
- 105 to avoid confusion please use 15 Oct 2018 and 31 Aug 2019.
Au: Was changed according the reviewer’s recommendation (see line 113).
- 305 Please explain more either here or in the discussion why there was a significant decrease only on two days. Do the co-authors think this is due to vaccination or are there other factors that could explain the decrease? Why would the reduction be less in group 1 and more in the others?
Au: Explanation is provided in lines 407-13
- 432 "East Asia" instead of "far East"
Au: Was changed accordingly (see line 438).
- 433-434 This concluding statement is a bit weak. "the data show" instead of "we believe". "without reservations" instead of "by all means".
Au: We thank the reviewer for this comment. The statement was changed accordingly (see lines 439-40).
- 435 Include the appendix as Supplemental Material or delete the Supplementary Materials statement.
Au: the appendix is now included in the supplementary material.
Reviewer 2 Report
The manuscript by Morgenstern and Klement present quantifiable data on the effect of vaccination of LSD live attenuated vaccine on milk production in cattle. The paper is very well written and present mathematical models properly designed to measure milk production, mortality and culling at different times post vaccination. Certainly, such information is of great scientific value given that outbreaks of LSD can severely impact the economy of countries that report such disease. One aspect of this study that needs clarification and proper data collection is to demonstrate that vaccination using live attenuated virus induced immunity against LSDV. This is an important point that serves as a control to fully correlate that vaccination does not impact milk production. Given that this study was conducted using live attenuated vaccines, a level of viral replication must have been obtained in the animal in order to provide immunity. This is not discussed or mentioned in the manuscript. The authors also should indicate the vaccination program that was used in the 73 farms that vaccinated cattle for 5 years vs the other 4 farms that only vaccinated for two years (2016 and 2019). Although, the authors mention that none of the farms tested were affected by LSD, they did not indicate what tests were conducted and whether viremias were examined by molecular testing (PCR) since animals suffering from LSD can be sub-clinically infected.
Minor comments:
- Line 18, remove “of”
- In the introduction, the authors should provide a brief introduction in the virus that causes the disease.
- A better description of the live attenuated vaccine used in this study should be provided. What kind of attenuation was done to the virus? Does the vaccine contain specific markers to differentiate vaccinated vs non-vaccinated?
- Line 69, it is not clear which vaccine was utilized to the study. It will be better to remind the reader that the homologous vaccine on attenuated Neethling LSD strain was used.
- Line 74, remove "a" data.
- Line 94, the reference indicated did not provide clear information on the nature of the live attenuated vaccine.
- Line 98, insert a space after "age"
- Supplementary materials were not accessible.
Author Response
The manuscript by Morgenstern and Klement present quantifiable data on the effect of vaccination of LSD live attenuated vaccine on milk production in cattle. The paper is very well written and present mathematical models properly designed to measure milk production, mortality and culling at different times post vaccination. Certainly, such information is of great scientific value given that outbreaks of LSD can severely impact the economy of countries that report such disease.
Au: We thank the reviewer for his support.
One aspect of this study that needs clarification and proper data collection is to demonstrate that vaccination using live attenuated virus induced immunity against LSDV. This is an important point that serves as a control to fully correlate that vaccination does not impact milk production. Given that this study was conducted using live attenuated vaccines, a level of viral replication must have been obtained in the animal in order to provide immunity. This is not discussed or mentioned in the manuscript.
Au: We agree with the reviewer. Indeed, provision of such data would have been very helpful. However, due to the retrospective nature of this study we were not able to test the elicitation of immune response by the specific vaccine lots used during this study. Nevertheless, there is no doubt that this vaccine elicits protective immune response as its efficacy and effectiveness was demonstrated and thoroughly analyzed in the past (see Ben Gera et al. 2015, and Klement et al. 2018). Evidence of the immunity elicited by the used batches can be provided by the fact that during a large epidemic which occurred in Israel shortly after this study no clinical cases of LSD appeared in vaccinated cows. We now discuss this in the manuscript (Lines 370-2, 417-20).
The authors also should indicate the vaccination program that was used in the 73 farms that vaccinated cattle for 5 years vs the other 4 farms that only vaccinated for two years (2016 and 2019).
Au: Data on vaccination program in all the farms is now provided in lines 97-106.
Although, the authors mention that none of the farms tested were affected by LSD, they did not indicate what tests were conducted and whether viremias were examined by molecular testing (PCR) since animals suffering from LSD can be sub-clinically infected.
Au: It is indeed true that subclinical infection by LSDV can occur. However, the occurrence of only subclinical infection in a farm with no clinical cases is highly implausible. Since the clinical signs of LSD are very prominent, we are sure that such infection could not go un-noticed in these farms.
Minor comments:
Line 18, remove “of”
Au: Was changed accordingly.
In the introduction, the authors should provide a brief introduction in the virus that causes the disease.
Au: Such information is provided in lines 32-4.
A better description of the live attenuated vaccine used in this study should be provided. What kind of attenuation was done to the virus? Does the vaccine contain specific markers to differentiate vaccinated vs non-vaccinated?
Au: A better description of the attenuated virus is now provided in the introduction (lines 48-55) and the materials and methods section (lines 102-6).
Line 69, it is not clear which vaccine was utilized to the study. It will be better to remind the reader that the homologous vaccine on attenuated Neethling LSD strain was used.
Au: See our reply to the previous comment.
Line 74, remove "a" data.
Au: Was changed accordingly.
Line 94, the reference indicated did not provide clear information on the nature of the live attenuated vaccine.
Au: we now provide a better reference for this information (lines 48-9)
Line 98, insert a space after "age"
Au: Was changed accordingly.
Supplementary materials were not accessible.
Au: the appendix is the supplementary material. We apologize for this mis-understanding. R-script and data will be provided through a link after acceptance for publication.
Round 2
Reviewer 2 Report
The authors have satisfactorily addressed many of my previous concerns. In addition, they have clarified or discussed remaining issues. The manuscript is significantly improved.